# Bacterial Small RNAs in the Genus *Herbaspirillum* spp.

**DOI:** 10.3390/ijms20010046

**Published:** 2018-12-22

**Authors:** Amanda Carvalho Garcia, Vera Lúcia Pereira dos Santos, Teresa Cristina Santos Cavalcanti, Luiz Martins Collaço, Hans Graf

**Affiliations:** 1Department of Internal Medicine, Federal University of Paraná, Curitiba 80.060-240, Brazil; santosvlp@hotmail.com (V.L.P.d.S.); hansgraf@bighost.com.br (H.G.); 2Department of Pathology, Federal University of Paraná, PR, Curitiba 80.060-240, Brazil; tecava@yahoo.com.br (T.C.S.C.); lmcollaco@uol.com.br (L.M.C.)

**Keywords:** RNA non-coding (ncRNA), ncRNA *cis*-encoded, *trans*-encoded, riboswitches, CRISPR, mRNA

## Abstract

The genus *Herbaspirillum* includes several strains isolated from different grasses. The identification of non-coding RNAs (ncRNAs) in the genus *Herbaspirillum* is an important stage studying the interaction of these molecules and the way they modulate physiological responses of different mechanisms, through RNA–RNA interaction or RNA–protein interaction. This interaction with their target occurs through the perfect pairing of short sequences (*cis*-encoded ncRNAs) or by the partial pairing of short sequences (*trans*-encoded ncRNAs). However, the companion Hfq can stabilize interactions in the *trans*-acting class. In addition, there are Riboswitches, located at the 5′ end of mRNA and less often at the 3′ end, which respond to environmental signals, high temperatures, or small binder molecules. Recently, CRISPR (clustered regularly interspaced palindromic repeats), in prokaryotes, have been described that consist of serial repeats of base sequences (spacer DNA) resulting from a previous exposure to exogenous plasmids or bacteriophages. We identified 285 ncRNAs in *Herbaspirillum seropedicae* (*H. seropedicae*) SmR1, expressed in different experimental conditions of RNA-seq material, classified as *cis*-encoded ncRNAs or *trans*-encoded ncRNAs and detected RNA *riboswitch* domains and CRISPR sequences. The results provide a better understanding of the participation of this type of RNA in the regulation of the metabolism of bacteria of the genus *Herbaspirillum* spp.

## 1. Introduction

The genus *Herbaspirillum* was described by Baldani et al. [1] and includes the *Herbaspirillum seropedicae* (*H. seropedicae*) species with several strains isolated from different grasses [1]. *Herbaspirillum seropedicae* is characterized by fixing nitrogen (diazotrophic) and colonizing plants.

A second diazotrophic species, *Pseudomonas rubrisubalbicans,* was reclassified under the name of *Herbaspirillum rubrisubalbicans* and also has the ability to colonize plants [2]. From the 1990s, new species were included in the genus *Herbaspirillum*: *H. autotrophicum* (*syn. Aquaspirillum autotrophicum*) and *H. huttiense* (*syn. Pseudomonas huttiense*) [3] *H. frisingense* [4], *H. lusitanum* [5], *H. cholorophenolicum* [6], *H. hiltneri* [7], *H. rhizosphaerae* [8], *H. aquaticum* [9], *H. massiliense* [10], *H. canariense*, *H. aurantiacum,* and *H. soli* [11], and *H. seropedicae* AU14040 [12]. The *H. putei* species was reclassified as *H. huttiense* subspecies *putei* [9].

*H. seropedicae* belongs to the beta class of Proteobacteria, is gram-negative, and displays a characteristic growth pattern in semi-solid culture medium and without nitrogen [1,13]. It is a bacterium that colonizes intracellular spaces and the xylem of grass roots, such as sugarcane and sorghum [1,14]; and intracellular spaces in seeds and rice roots [14,15]. It can also be isolated from the stem leaves of various grasses [15,16,17]. The genome of the *H. seropedicae* strain (SmR1) has been completely sequenced and annotated by the GENOPAR (Paraná Genome) consortium. “Wordiness 4,735” Open Reading frames (ORFs) occupy about 88.3% of the genome, on a single circular chromosome.

In this annotation, the genes of non-coding RNAs (apart from rRNAs and tRNAs), are not included. Gene encoding regulatory RNAs, known as small RNAs (sRNAs) or non-coding sRNAs (ncRNAs), are transcribed in *trans* and in *cis* relative to target RNA [18,19]. The work of the researchers Anderson et al. were the first to isolate and characterize the RNA encoded by the micF gene in *Escherichia coli* [20].

The genes are located between the coding regions for the proteins, that is, in the intergenic regions of the genome and display signals from the promoters and terminator sequences that are usually Rho-independent [21,22,23]. The size of the ncRNAs genes ranges from ~50 to ~500 nucleotides and several transcripts were processed by RNase into smaller products [24,25,26,27]. They modulate physiological responses through different mechanisms, by RNA-RNA interaction or RNA-protein interactions, and some interactions can be regulated by the Hfq chaperone [28,29,30].

The most studied ncRNAs are the *cis*-encoded RNAs and the *trans*-encoded RNAs. The *cis*-encoded RNAs are transcribed in *cis* and *encoded* at the same locus as their target mRNA in the antisense sense duplex structure, in addition to having a post-transcriptional gene regulation of responses and a high degree of sequence complementarity, which was considered to be an indication that interaction of the Hfq protein [31,32] would not be required. However, some investigators reported interference of Hfq in the ncRNA, mRNA targeting, and *cis*-encoded target [33,34,35]. Overall, these ncRNAs act by complementing the mRNA ribosome-binding site, as well as inhibiting the conversion [33,36].

In contrast, *trans*-encoded ncRNAs that are encoded in *trans*, have target mRNAs at different sites in the genome where the formation of the ncRNA occurs: the Hfq—mRNA complex may act positively or negatively on post-transcriptional regulation [37]. This is due to the imperfect base pairing which prevents their eventual degradation by RNase [38,39]. The regulation of gene expression of *trans*-encoded ncRNAs was first discovered by Dalihas, who described the micF and dicF found in *Escherichia coli*, and the lin-4 found in the nematode *Caenorhabditis elegans*, which are encoded by genes located at different sites of their target genes and only display a partial complementarity with their target RNAs [40].

The *Riboswitches*, which are located in the 5′ UTR region of an mRNA, constitute another class of ncRNAs and lead to transcriptional regulation through their interaction with a linker molecule [41,42,43]. However, the presence of small, non-coding RNAs in different *Herbaspirillum* species remains undetermined. A total number of 285 ncRANs are needed to predict the novel candidate sRNAs within the intergenic regions (*Herbaspirillum seropedicae* SmR1). We further analyzed the predicted ncRNAs in great detail in *Herbaspirillum* spp. using Infernal 1.1.1 bioinformatic tools. Moreover, these analyses constitute the very first documented evidence of the presence of sRNAs in *Herbaspirillum* genomes.

## 2. Results

### 2.1. Prediction of New ncRNAs in the Genus Herbaspirillum spp.

The computational tool Infernal 1.1.1 was employed for this study (see Section 4.3) and the multiple covariance detection and sequence analysis had as its input the RNA family database (Rfam, available link http://rfam.xfam.org/; accessed on 18 February 2016), while the prediction of the 16 species of ncRNAs made up the genus *Herbaspirillum* spp., as shown in Appendix A.

The complete list of genomes is as follows: *H. seropedicae* Z67, *H. lusitanum* P6-12, *H. hiltneri* N3, *H. frisingense* GSF30, *Herbaspirillum* spp. strain B65, *H. rubrisubalbicans* M1, *H. autotrophicum* IAM 14942, *H. huttiense* subsp. *putei*, *Herbaspirillum* spp. strain B501, *Herbaspirillum* spp. strain GW103, *Herbaspirillum* spp. strain RV423, *H. rhizosphaerae* UMS-37, *Herbaspirillum* spp. strain Os34, *Herbaspirillum* spp. strain Os45, and *H. seropedicae* AU14040. The results are listed in Table 1. By adopting this approach, no ncRNAs were detected in *H. lusitanum* P6-12 and *Herbaspirillum* spp. strain B65, due to the significant number of scaffolds and contigs that are identified in incomplete genomes. The organisms with the highest number of common ncRNAs were *H. seropedicae* SmR1, *H. seropedicae* Z67, *H. hiltneri* N3, *H. frisingense*, *H. rubrisubalbicans* M1, and *H. seropedicae* AU14040, shown in Appendix A.

The ctRNA_p42d, Betaproteobacteria_toxic_sRNA, and P10 ncRNAs are examples of ncRNAs found in *Herbaspirillum* spp. RV423, *H. rhizosphaerae* UMS-37, and *Herbaspirillum* spp. OS34 and O45 and common to *H. seropedicae* SmR1. The ncRNAs *Betaproteobacteria_toxic_sRNA*, *pfl*, *SX4*, and *alpha_tmRNA*, *AR35*, *tmRNA*, among others, are repeated in all the species, except for *Herbaspirillum* spp. B65, *Herbaspirillum* spp. B501, and *Herbaspirillum lusitanum* P6-12. The characterization of the ncRNAs predicted in the genus *Herbaspirillum* spp. suggests that despite the differences within these species, there is sequence preservation among the common ncRNAs.

In *Herbaspirillum seropedicae* SmR1, the prediction of ncRNAs resulted in 108 ncRNAs, as shown in Figure 1. The sequence of ncRNAs predicted in *H. seropedicae* SmR1 by Infernal 1.1.1 is shown in Appendix A.

All ncRNAs predicted by the Infernal 1.1.1 tool in the genus *Herbaspirillum* spp. are shown in Appendix A.

### 2.2. ncRNAs Differential Expression in RNA-Seq

Data from RNA-seq data were used in 11 culture conditions that were tested in *Herbaspirillum seropedicae* SmR1, as described in Appendix A. Less than five covering values were included and these were regarded as possible predicted ncRNAs. It was observed that there were ncRNAs expressed in all 11 conditions, but others were only expressed in some of the conditions.

The 5′_ureB_sRNA in *H. seropedicae* SmR1 has 286 nucleotides and is classified as *cis*-encoded antisense ncRNA depending on its function and localization of the genome, is incorporated with the *ureA, ureB, ureC, ureD, ureE,* and *ureF* (sense) genes, is expression the NFbHPN-Malate-Naringenin and NFbHP-Malate-2 conditions, as shown in Figure 2. *H. seropedicae* SmR1 uses naringenin as a carbon source and is found in the plant, where the biosynthesis of flavonoids occurs; its compounds are involved in the plant defense mechanism.

The ncRNA 5′_ureB_sRNA is repeated in some genera of *Herbaspirillum* by having 100% sequence identity such as, for example, in *H. seropedicae* Z67, *H. lusitanum*, *H. hiltneri* N3, *H. rubrisubalbicans* M1, and *H. seropedicae* AU14040. The location of the ureABC operon locus in the genome of these species was identified. The presence of ureaseC (alpha), ureaseA (gamma), and ureaseB (beta) subunits was detected and it was found that the 5′-ureaB sRNA ncRNAs were found in these species of *Herbaspirillum* interacts in *cis* of the ureaseC subunit. The ncRNA mRNA 5′ UTR cspA is a thermoregulator which has 373 nucleotides in *H. seropedicae* SmR1. It is classified as a *cis*-encoded ncRNA, which appears as a mRNA 5′ mRNA sequence of the gene annotated as *cspD* (*Hsero_1397*). In addition, there is a second *cspD* gene (*Hsero_3028*) that does not contain the mRNA sequence 5′ UTR.

Yamanaka and Inouye [44] used the cspD-lacZ fusion and found that the cspD expression induced by stationary phase growth does not depend on sigma factor 6S. Thus, the *cspD* gene expression is inversely dependent on the growth, rates induced by glucose starvation. They also observed that the 5′ UTR cspA mRNA is expressed in the all assay conditions and showed a higher expression in seq RNA assays at NFbHPN-Malate-Nitrate and NFbHP-Malate-Nitrate-2 and NFbHPN-Malate-High-Oxygen-1, as shown in Figure 3. NFbHPN-Malate-Nitrate-1 is where *H. seropedicae* SmR1 rises to 30 °C in NFbHP medium, using malate as a carbon source and 10 mM nitrate as a nitrogen source [45].

The ncRNA sX4 has 113 nucleotides and is classified as *trans*-encoded antisense ncRNA. It is co-located with the *ampG*, *tufB*, and *trnG* genes. The designation sX4 refers to the ncRNAs described in *Xanthomonas campestris pv. vesicatoria* [46]. In *H. seropedicae* SmR1, the sX4 ncRNA was expressed in all the tested conditions, but the highest expression in Adhered-corn-3-days conditions, followed by the Plankton-corn-3-days-2. It was found that the greatest expression occurred during the colonization of maize, as shown in Figure 4. Thus, it should be noted that the greatest expression occurred during the colonization of corn.

In *H. seropedicae* the *cobalamin riboswitch* has 113 nucleotides and is classified as *trans*-encoded antisense ncRNA or cobalamin element. It is co-located with the *Hsero_2659* gene (cobalt transporter) as a sequence 5′ UTR, and with the genes *Hsero_2660* and *Hsero_2661* (antisense sense) genes. Coenzyme B12 or cobalamin riboswitches are control elements that are widespread in prokaryotes. For example, the *metE* gene contains a cobalamin *riboswitch* and binding of coenzyme B12 to this element that leads to the formation of a transcriptional terminator and the repression of *metE* expression. In the absence of the coenzyme, the *metE* mRNA is synthesized [47]. NFbHPN-Malate-Low-Oxygen-1 and NFbHPN-Malate-Naringenin-1 are shown in Figure 5. These results indicate that there is likely to be no binding of the CbtB (cobalt carrier subunit) linker to the *hsero_2659* mRNA in the aptamer region, where an alternate clamp loader formed.

Non-coding RNAs were regarded as essential (*housekeeping* ncRNAs), such as 6S ncRNA, and were also identified by the Infernal 1.1.1 tool. In *Herbaspirillum seropedicae* SmR1, mcRNA 6S has a sequence of 177 nucleotides and expression in RNA-seq material. The ncRNA 6S in *E. coli* is responsible for regulating the activity of the RNA polymerase (RNAP) that is dependent on the sigma factor 70, and necessary for the recognition of the promoter and initiation of the transcription. In addition, it globally regulates the gene expression response to growth from the exponential phase to the stationary phase [48]. It is significant that the level of expression was found to be high relative to its neighbors (*Hsero_1224* and *Hsero_1225*) under the conditions tested, and this confirms that this ncRNA is essential. In addition, the 6S ncRNA was expressed in all the conditions tested, with an emphasis on the Adhered-wheat-1 condition, as shown in Figure 6.

### 2.3. Comparative Analysis of the Secondary Structure of 6S of H. seropedicae SmR1

Given the importance of this ncRNA, there was a comparative analysis of the secondary structure of the 6S of *H. seropedicae* SmR1, together with the 6S of *Janthinobacterium* spp. and *Burkholderia* CCGE1003. The analysis was conducted with the RNAFold [49] tool, and based on thermodynamic RNA-RNA interactions, where the secondary structures of nCRNA 6S were generated. The three structures shown in Figure 7 are basically similar (a long rod shown in Figure 7B for the 6S of *Janthinobacterium* spp.). However, there are variations in the presence and number of loop out structures that may or may not have a functional meaning. A multiple sequence alignment was carried out by the CLUSTALW tool [50], as shown in Figure 8. It is observed that there is a difference in the secondary structure of ncRNA 6S between these species, even if they present a high degree of conservation in the aligned nucleotide sequence.

### 2.4. Candidate sRNA Target Identification

In *H. seropedicae* SmR1, the consensus ribosomal binding site (RBS)sequence and its variations were examined, since this is the RBS AGGA nucleus and the CAAGGACA sequence. Take as an example the nucleotide sequence of ncHSmR1_04, which has 40 possible targets. They include the *serS* gene that encodes a seryl-tRNA synthetase.

It is worth noting that the base pairing between the ncHSmR1_04 and the mRNA serS, occurs in the RBS region of the latter, as shown in Figure 9. This is a strong indication that the ncHSmR1_04 negatively regulate the conversion of seryl-tRNA synthetase. The 4.5S RNA is a common ncRNA in bacteria and its purpose is to the function of directing proteins containing signal peptide to the secretory apparatus, and thus form a signal-recognizing particle with the Fhh protein. In addition, the 4.5S RNA takes part in the conversation when interacting with the G stretch factor of G [51].

In *E. coli*, the 4.5S RNA is encoded by the gene (*ffs*) and contains 114 nucleotides. It is the bifunctional molecule that is involved in the conversion and secretion of protein by binding to elongation factor (EFG) and Ffh the protein respectively [52]. Apparently, the 4.5S RNA facilitates the release of the EF-G-GTP from the ribosome by competing with the 23S rRNA for binding with the EF-G [51,53].

Buskiewicz et al. [54] analyzed the functions of the SRP (signal-recognizing particle) in *E. coli* that were based on the binding of the Fth protein to the 4.5 S RNA and the Fth-4.5 S complex. The researchers suggested that the free Fth, the 4.5S RNA, and FtsY binding sites are occluded by strong domain–domain interactions that must be disrupted by the SRP or Fth-FtsY complex formation. 

The 4.5S RNA of *H. seropedicae* SmR1 contains 146 nucleotides. Among the targets of 4.5S, as predicted by Target RNA2, is the *mutL* mRNA that encodes a protein involved in DNA repair. Again, the matching of the ncRNA occurs close to that of the RBS of the target RNA and in this specific case, starts at the third base upstream of the adenosine of the AUG codon, as shown in Figure 10.

The analysis of the ncRNA 4.5S conducted by the TargetRNA2 tool resulted in 77 target mRNAs in this set and includes the *nifQ* gene whose product is involved in nitrogen metabolism, as shown in Figure 11.

It can be seen that the predicted pairing starts 21 bases upstream of the AUG codon, and hence is outside of the RBS region, as shown in Figure 12. This means that according to TargetRNA2, when compared with the previous example, an interaction energy value is obtained for the pairing: −16.21 for *mutL* and −8.31 for *nifQ*. These two factors (the pairing position and interaction energy value) make *nifQ* an unlikely target for 4.5S.

### 2.5. Prediction of CRISPR

A new category of non-coding RNA, CRISPR RNA, has been described for both the Bacteria domain and the Archaea domain [55,56]. The CRISPR transcription factor results in small fragments of RNA that recognize a specific exogenous DNA and that can guide the Cas nuclease, that is responsible for the cleavage of this DNA if it re-contacts the microorganism. Thus, prokaryotes have a defense mechanism against invasive DNAs [57]. In this study, the CRISPRFinder and CRISPRmap tools were used to predict CRISPR in the genome of *H. seropedicae* SmR1. A CRISPR locus was identified with 164 nucleotides divided into DR (54 nucleotides) and SPA (57 nucleotides) palindromic repetitions, as shown in Figure 12.

The CRISPR locus is co-located with the *rhtB* gene (sense-sense) and with the *psiF* and *Hsero_1878* genes (in the antisense-sense), as shown in Figure 13. With regard to this, CRISPR is expressed in some conditions, with the highest level being expression in the condition NFbHPN-Malato-Alto-Oxygen-1 a condition where *H. seropedicae* SmR1 is using malate as a carbon source up to OD of 0.4. This result suggests that the expression of the CRISPR locus in the NFbHPN-Malate-High-Oxygen-1 condition acts as a signal for the nitrogen source that the bacterium is using, by assisting in controlling the pH of the medium, since the *psiF* gene is expressed during the phosphate deprivation.

In addition, the CRISPRFinder tool identified the CRISPR-associated protein, while the Cmr4 Family RAMP in *H. seropedicae* SmR1 interacted with casRAMP_Cmr4, which participates in the process of developing immunological e memory. The Cas RAMP complex modulates between the cleavage of the RNAs [58,59,60,61]. The results suggest that the CRISPR-Cas system is a prokaryotic defense mechanism against foreign genetic elements. The main features of this defense system are Cas proteins and CRISPR RNA. However, since the different nitrogen sources have variations in the amount of nitrogen and CaCo3 equivalents, acidification and the reduction of soil pH can become significant.

## 3. Discussion

In the RNA-seq assays, the highest expression of 5_ureB_sRNA ncRNA in the NFbHPN-Malate-Naringenin condition was observed, which may suggest that this 5_ureB_sRNA ncRNA is involved in the regulation of flavonoid biosynthesis. These act as a protection against oxidizing agents in the plant and involve the use of externally generated and internally generated urea as a source of carbon. Urea is metabolized by urease, a nickel-dependent metalloenzyme, which catalyzes the hydrolysis of urea to form ammonia and carbon dioxide, and can be found in bacteria, fungi, and plants. [62].

In *Klebsiella aerogenes*, the ureABC operon encodes the three subunits of the pimple enzyme [62]. In *Helicobacter pylori*, the urease cluster consists of two operons (ureAB and ureIEFGH). This bacterium is the only one described to date that contains the 5_ureB_sRNA, an antisense *cis*-encoded ncRNA with 290 nucleotides, which negatively regulates ureAB expression when paired at the 5′-region of the ureB mRNA [63]. *H. seropedicae* SmR1 has a probable ureABC operon and a 5′_ureB ncRNA, with 286 nucleotides. According to the prediction of Infernal 1.1.1, this ncRNA is a *cis*-encoded antisense that is positioned close to the ureC gene. This difference can be explained by the fact that (a) *H. pylori* does not have the ureC gene, (b) the *H. seropedicae* SmR1 gene sequences are larger than those of *H. pylori,* and (c) the metabolism of urea involves different proteins.

According to Yamanaka and Inouye [44], in *E. coli* there are new homologues of cspA, denominated cspA to cspI, and the cspD gene does not have the 5′ UTR sequence. This information suggests that in *H. seropedicae* SmR1, the genes cspD (*Hsero_1397*) is actually the cspA gene. This gene is co-located with the genes *rsuA, icd, clpS, clpA* (sense sense). Jiang et al. [43] analyzed the function of the unconverted 5′ UTR sequence CspA mRNA from the major cold shock adaptation protein to CspA in *E. coli*. The researchers observed that adaptation to cold shock is blocked when the 5′ UTR CspA mRNA region is overexpressed, through the synthesis not only of the CspA protein, but also of proteins from the CspA family such as CspB, CspC, CspD, and CspE. The cspD gene in *E. coli* codes for a protein of homologous sequence with the cold shock protein to the CspA, but the expression of cspD is not induced by the cold shock [44].

The sX4 ncRNA whose expression is dependent on the HrpG and HrpX proteins, suggests that the target (s) of this ncRNA can be found in the interaction of the bacterium with the plant [46,64]. Studies with *HrpG* and *HrpX* mutants in *X. campestris pv. vesicatoria* reveal that these genes are essential for pathogenicity and assist in the survival of epiphytic bacteria [65]. These microorganisms are gram-negative and pathogenic γ-proteobacteria of many plant species of economic interest.

The ncRNA MicC has 97 nucleotides, is classified as *trans*-encoded antisense, and is co-located with the *murI* and *smoked* genes and with the *gst* gene (sense). In *E. coli*, the MicC ncRNA is located between the *ompN* and *ydbk* genes, and regulates OmpC protein expression by pairing with the mRNA leader sequence and inhibiting ompC mRNA binding to the ribosome [47,65].

In the genome of *H. seropedicae* SmR1, the *ompC* gene is located at the *Hsero_1282* locus, the *ompA* gene is located at the *Hsero_1287* locus and the *ompR* gene is located at the *Hsero_1518* locus. With the aid of the TargetRNA2 tool, it can be observed that the MicC ncRNA in *H. seropedicae* SmR1 targets the rscB gene located at the Hsero_1538 locus, which acts as a transcriptional regulator that has a receptor domain for the OmpR protein. The ncRNA MicC was expressed in all the tested conditions and achieved higher expression observed in Adhered-corn-3-days-2 and Aderido-maize-3-days-1 conditions, with a higher level of expression with regard to the other conditions. In addition, it is involved in the transcriptional expression of the rcsB gene when it is adhered to maize.

In *H. seropedicae* SmR1, the IsrD ncRNA has 62 nucleotides, and is co-located with the *mmr* and *Hsero_0991* genes, together with the *Hsero_0992* gene (antisense sense). It has the highest expression profile in RNA-seq material under Planktonic-corn-3 days-2 and NFbHPN Malate-High-Oxygen-2 conditions. The IsrG ncRNA has 88 nucleotides, is co-located with the *fimV* and *asd* genes, and with Hs_noco_697 (sense sense), and is only expressed in the NFbHPN Malate-High Oxygen condition. In *Salmonella typhimurium* ncRNA, IsrG is involved in the response to cold shock and acid pH, in the stationary phase of growth [66]. According to the literature, genes from the ncRNA-isr family have a promoter sequence superimposed on the 5′ end or the 3′ end of the neighboring gene [67]. In *H. seropedicae* SmR1, it was observed that the IsrD gene is superimposed on the 3′ end of the *Hsero_0991* gene, already for IsrG, expressed in the NFbHPN Malate-High Oxygen condition; it was also observed that it overlaps the *fimV* gene.

The TargetRNA2 computational tool was used to predict ncRNA targets in *H. seropedicae* SmR1. This program was specifically designed for the prediction of mRNA targets of ncRNAs with activity in *trans*. As a result, a wide range of mRNA targets were observed, regardless of the ncRNA that was analyzed, because after transcription the *cis*-encoded ncRNA make short and perfect pairing with their targets and the *trans*-encoded ncRNAs make long and imperfect pairing [68].

The number of targets for each ncRNA was variable. For example, ncHSmR1_02 had 11 targets, ncHSmR1_33 had 45 targets, and ncHSmR1_23 with 99 targets. In the case of some ncRNAs, such as ncHSmR1_01, there was no prediction of targets. The mRNA targets predicted in *H. seropedicae* SmR1 triggered by different biochemical mechanisms, for example, some of the targets of ncRNA 4.5S are as follows: argB (acetylglutamate kinase), *Hsero_3355* (lipoprotein), *glk* (glucokinase), *htpX* (heat shock protein HtpX), *Hsero_2759* (transcriptional regulatory protein).

The ncHSmR1_45 with 11 targets, ncHSmR1_149 with 45 targets, and ncHSmR1_104 with 99 targets, ncHSmR1_01 with 16 targets, ncHSmR1_140 with 5 targets. Moreover, in the case of some non-coding RNAs, such as ncHSmR1_42, there was no prediction of targets. This was probably because the mRNA sequences deposited in the TargetRNA2 tool, did not match ncHSmR1_42. The mRNA targets predicted in *H. seropedicae* SmR1 are related to different biochemical mechanisms, for example, among the 4.5S RNA targets are: argB (acetylglutamate kinase), Hsero_3355 (lipoprotein), glk (glucokinase), htpX and HtpX (Heat shock protein), and Hsero_2759 (transcriptional regulatory protein). These results should lead to a better understanding of the participation of this type of RNA in the regulation of the metabolism of *Herbaspirillum seropedicae* SmR1.

Among the mRNA ratios for the predicted targets, an analysis was conducted of the likely target ncRNA-RNA interaction site, with an emphasis on those cases in which this interaction occurs near or over the RBS region. The first experimental evidence for the binding of an ncRNA to the ribosomal binding site of a messenger RNA and blocking the binding site, as a result of the base pairing between micF antisense RNA and the target mFNA mRNA, has been described in *Escherichia coli* in the final 5′ regions of the latter [69].

Three ncRNAs were identified in *H. seropedicae* SmR1, with expression observed in RNA-seq material. The results indicate the existence of *cis*-encoded ncRNA, *trans*-encoded ncRNA, *riboswitch,* and CRISPR in *H. seropedicae* SmR1. Micc, IsrK, IsrD, and IsrG Hfq-dependent ncRNAs were detected among these. The number of target mRNAs predicted by the TargetRNA2 tool for the *H. seropedicae* SmR1 ncRNAs varied considerably and there was also a variation in the position of the mating region in relation to the AUG codon. The Infernal 1.1.1 tool did not detect ncRNAs in *H lusitanum* P6-12 and *Herbaspirillum* sp. B65.

## 4. Materials and Methods

### 4.1. Genomes of Bacteria of the Genus Herbaspirillum

The genomes of *H. seropedicae* SmR1, *H. seropedicae* Z67, *H. lusitanum* P6-12, *H. hiltneri*, *H. frisingense* GSF30 N3, *Herbaspirillum* spp. strain B65, *H. rubrisubalbicans* M1, *H. autotrophicum* IAM 14942, *H. huttiense* subsp. *putei*, *Herbaspirillum* spp. strain B501, *Herbaspirillum* spp. strain GW103, *Herbaspirillum* spp. strain RV1423, *H. rhizosphaerae* UMS-37, *H. rubrisubalbicans* spp. strain Os34, *H. rubrisubalbicans* spp. strain Os45, and Herbaspirillum AU14040 were obtained from the database of the National Center for Biotechnology Information (NCBI) search. Species of Herbaspirillum, with access codes and date described, are in Appendix A.

### 4.2. RNA-Seq Data

The researchers of the Biological Nitrogen Fixation (BNF) Centre kindly provided gene expression data of *H. seropedicae* SmR1. The in-plant culture and/or inoculation conditions are described in Appendix A.

### 4.3. Prediction of ncRNAs in Silico

The *H. seropedicae* SmR1 genome was screened with the INFERence tools of RNA Alignment by employing a covariance model to show probabilistic profiles of sequences and secondary structures of RNA families [70]. The covariance model is a special case of a probabilistic mechanism that seeks to combine the consensus sequence with the secondary consensus structure of a given RNA. In many cases it is able to identify homologous RNAs that have a conserved secondary structure but low primary sequence conservation [71]. Infernal 1.1.1 available link http://eddylab.org/infernal/; accessed on 18 February 2016), consists of five programs: cmbuild, cmcalibrate, cmsearch, cmscan, and cmalign. The following command line was used: cmscan-o sequenceSMR1.out-tblout sequenceSMR1.tbl-T 24-notrunc Rfam.cm.sequenceSMR1 where sequenceSMR1, which represents the genome of the *H. seropedicae* SmR1 bacterium, has been replaced by the genomes listed in Section 4.1. 

### 4.4. ncRNA Sequence Annotation

The ncRNAs sequences were annotated using the Rfam online database. This database contains a collection of ncRNA families represented by sequences of manually edited alignments, consensus secondary structures, and annotations taken from taxonomic sources and ontology [70]. The base is a broad and diverse source of ncRNAs and includes 2791 families of ncRNAs, with information on various types of ncRNAs throughout the three life domains and viruses. Infernal 1.1.1 was used to make multiple sequence alignments by means of the covariance model. In addition to the annotation of ncRNAs, Rfam 14.0 classifies ncRNAs and provides bibliographical references for each family, links to the PDB (Protein Data Bank), ENA (European Nucleotide Archive), and Gene Ontology (GO). The ncRNA candidates were also compared with the online Ribe dataset [72] so as to identify putative *riboswitches*. This database is capable of recognizing conserved sequences and known *riboswitches* located upstream of orthologous genes in multiple genera [73]. Also, the ncRNA candidates were compared with the NCBI database using the BlastX/N webserver. 

### 4.5. Mapping and Visualization of Sequence Reads

We masked the rRNA sequences through a cross-match program and carried out a recursive read trimming at 5′ and 3′ to 35 nucleotides using a Perl script to map the short reads required for targeting the genome. The resulting Mate-Paired reads were aligned to the *H. seropedicae* SmR1 genome using the SHRiMP alignment tool [74]. The program was set up to tolerate three mismatches. We used SAMtools available link http://samtools.sourceforge.net/; accessed on 17 February 2016) [75] to convert date into the SAM/BAM format. Mapped RNA-seq reads in BAM format were visualized in the Artemis genome browser [76].

### 4.6. Transcriptome Analysis

The standardization of the samples sequenced on the SoliD platform was carried out by the RPKM (reads per kilobase of transcript per million mapped reads) method, which estimates the gene expression value of a gene by measuring the density of the readings in a gene region of interest, and standardizing the readings counts in their exonic regions, compared with the original size of the gene or exon [77].

### 4.7. Target Prediction

The TargetRNA2 prediction tool was employed to target mRNAs that base-base with ncRNAs [68,78]. It calculates the hybridization score and the statistical significance of interactions between mRNA-ncRNA [79]. The tool uses a wide range of features to identify the target mRNAs, among them the conservation of ncRNA. This feature makes a comparison of the available sequence in GenBank in terms of the deposited genome (replicon) and indication of regions that have a greater conservation of sequences that are prone to undergoing mRNA/cRNA interactions. Another feature is the accessibility structure of the mRNA mRNA/ncRNA, which is weighted by its stability and regions of interactions. In addition, account is taken of the energy hybridization regions of the mRNAs with a low energy index for hybridization in one or more regions of interactions with potential targets [68].

### 4.8. Identifying CRISPR

A computational tool to identify CRISPR classification repeat conservation was used. This allows the independent grouping and determination of conserved sequence families, structural motifs suitable for endoribonucleases, and evolutionary relations [80]. CRISPRFinder is an interface that provides a detailed analysis of CRISPR genomic sequences [81].

## 5. Conclusions

The method used by Inferna1.1.1. for prediction of ncRNAs in the genus *Herbaspirillum* spp. prioritizes the identity of the various classes of ncRNAs. This RNA alignment inference has perspective combining the biological premise on associations of DNA sequences to RNA structure and sequence similarities. At the same time incorporation of the covariance models (CMs) scores a combination of sequence consensus and secondary RNA structure consensus, therefore, ncRNAs in the genus *Herbaspirillum* spp. Were identified, given the importance of ncRNA expression in seq RNA in in the strain *Herbaspirillum seropedicae* SmR1. The results provide a better understanding of the mechanism of post-transcriptional regulation of ncRNAs due to the interaction of mRNAs mRNAs and the diversity of ncRNAs classification in the genus *Herbaspirillum* spp.

## Figures and Tables

**Figure 1 ijms-20-00046-f001:**
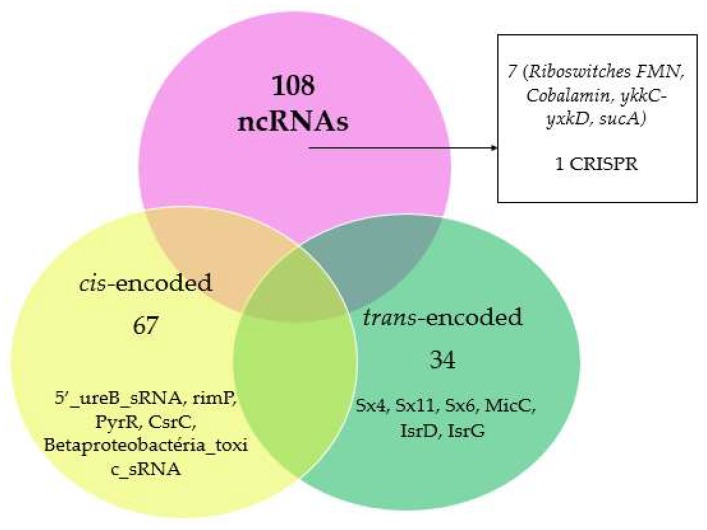
Examples of the *cis*-encoded 5′_ureB_sRNA, rimP, PyrR, CsrC, Betaproteobacteria_toxic_sRNA, riboswitch, FMN, cobalamin, ykkC-yxkD, sucA; and the *trans*-encoded sX4, sX11, sX6, MicC, IsrK, IsrD, IsrG. The CRISPR RNA region predicted the *H. seropedicae* in the SmR1 genome.

**Figure 2 ijms-20-00046-f002:**
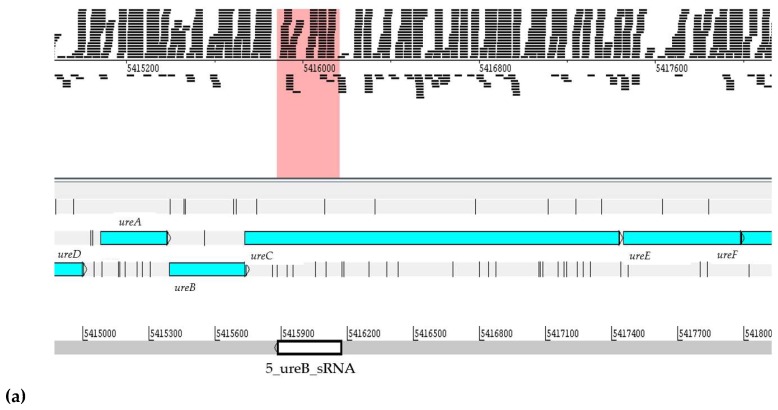
Visualization of position and expression profile of 5_ureB_sRNA ncRNA in the genome of *H. seropedicae* SmR1. The sequence was located using the Artemis program. The readings delineated within the pink rectangle refer to: (**a**) Levels of expression of 5_ureB_sRNA ncRNA in RNA-seq assays in *H. seropedicae* SmR1. In color, the analyzed experimental conditions represented are shown, where 1 and 2 represent d.a.i (days after inoculation); (**b**) the numbers represent the value of the gene expression in reads per kilobase of transcript per million mapped reads (RPKM).

**Figure 3 ijms-20-00046-f003:**
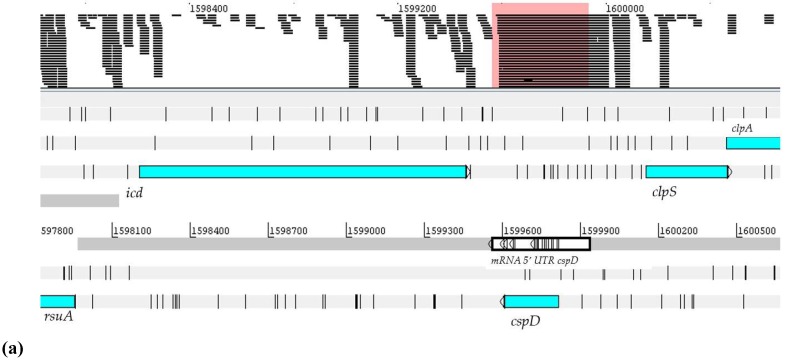
Visualization of the position and expression profile of mRNA 5′ UTR *cspA* in the genome of *H. seropedicae* SmR1. The sequence was located with the aid the Artemis program. The readings delineated within the pink rectangle refer to: (**a**) Levels of expression of mRNA 5′ UTR *cspA* in RNA-seq assays in *H. seropedicae* SmR1. In the colored part, the 11 analyzed experimental conditions are represented, where 1 and 2 represent d.a.i (days after inoculation); (**b**) the numbers represent the value of the gene expression in RPKM.

**Figure 4 ijms-20-00046-f004:**
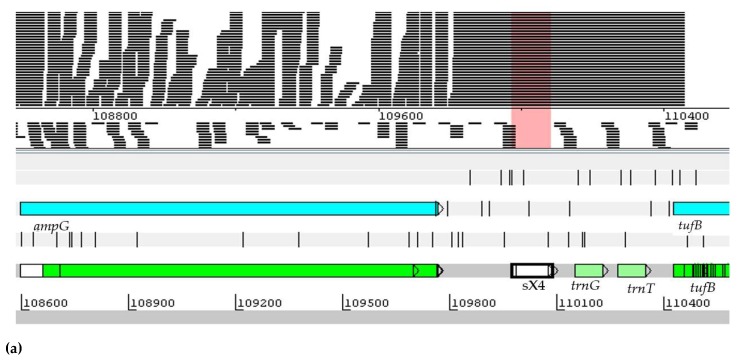
Visualization of the position and expression profile of the sX4 ncRNA in the genome of *H. seropedicae* SmR1. The sequence was located by means the Artemis program. The readings delineated by the pink rectangle are as follows: (**a**) Levels of sX4 ncRNA expression in RNA-seq assays in *H. seropedicae* SmR1. In the colored section, the 11 analyzed experimental conditions are represented, where 1 and 2 represent d.a.i (days after inoculation); (**b**) the numbers represent the value of the gene expression in RPKM.

**Figure 5 ijms-20-00046-f005:**
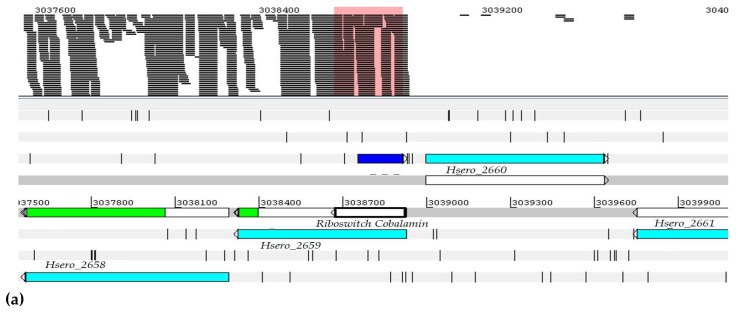
Visualization of the position and expression profile of the cobalamin ncRNA in the *H. seropedicae* genome SmR1. The sequence was located by means of the Artemis program. The readings are delineated by the pink rectangle (**a**). Levels of ncRNA cobalamin expression in RNA-seq assays in *H. seropedicae* SmR1. In the colored section, the 11 analyzed experimental conditions represented, where 1 and 2 represent d.a.i (days after inoculation). (**b**) The numbers represent the value of the gene expression in RPKM.

**Figure 6 ijms-20-00046-f006:**
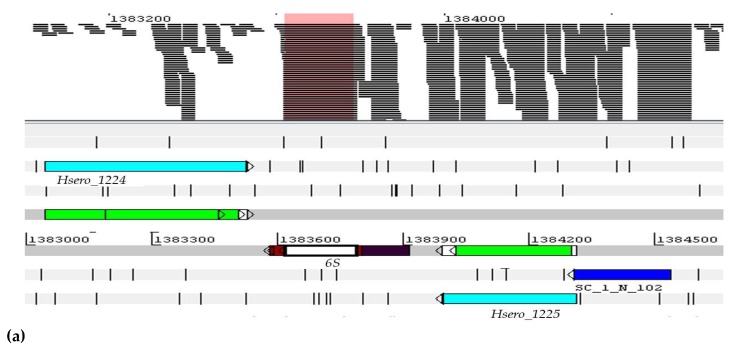
The sequence was located by means of the Artemis program. The readings delineated by the pink rectangle were as follows: (**a**) Visualization of expression levels of ncRNA 6S in RNA-seq assays in *H. seropedicae* SmR1. In the colored section, the 11 analyzed experimental conditions represented, where 1 and 2 show d.a.i (days after inoculation); (**b**) the numbers represent the value of the gene expression in RPKM.

**Figure 7 ijms-20-00046-f007:**
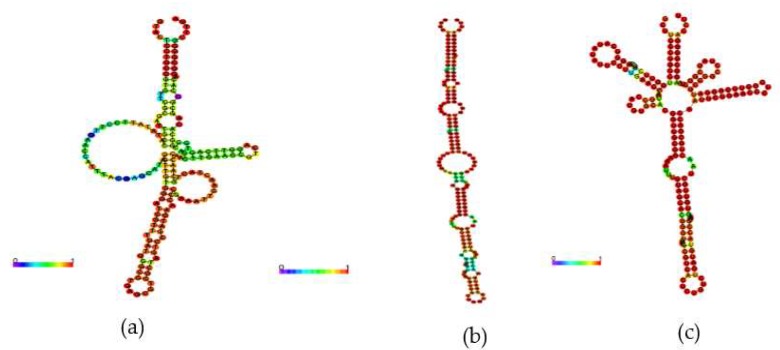
Visualization of the secondary structure of ncRNA 6S. (**a**) *H. seropedicae* SmR1; (**b**) Janthinobacterium sp.; (**c**) Burkholderia CCGE1003. A predictive analysis of the secondary structures of identified ncRNAs was conducted by means of RNAFold.

**Figure 8 ijms-20-00046-f008:**
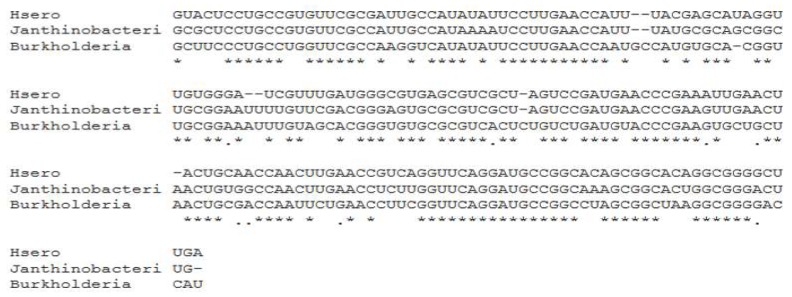
Multiple alignment of 6S ncRNA gene sequences by the CLUSTALW tool. Hsero = *Herbaspirillum seropedicae* SmR1; Janthinobacteri = *Janthinobacterium* sp.; and Burkholderia = *Burkholderia* CCGE1003.

**Figure 9 ijms-20-00046-f009:**
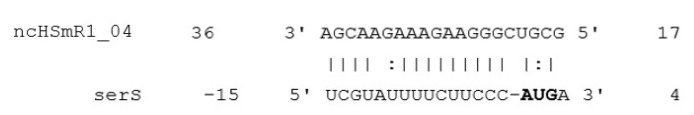
It is worth noting that the base pairing between the ncHSmR1_04 and the mRNA serS occurs in the RBS region of the latter.

**Figure 10 ijms-20-00046-f010:**
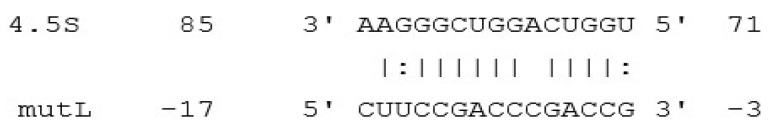
It is worth noting that the base pairing between the 4.5S and the mRNA mutL start at the third base upstream of the adenosine of the AUG codon.

**Figure 11 ijms-20-00046-f011:**
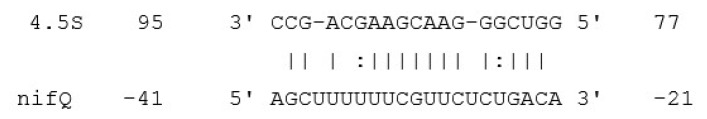
It is worth noting that the base pairing between the 4.5S and the mRNA *nifQ* predicted pairing starts 21 bases upstream of the AUG codon, and hence is outside of the RBS region.

**Figure 12 ijms-20-00046-f012:**
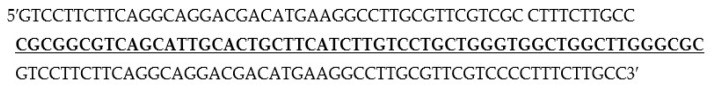
Visualization of the CRISPR locus in the genome of *H. seropedicae* SmR1. Sense sense in the genome, with 164 nucleotides divided into DR (54 nucleotides) palindromic repetitions and spacing region (57 nucleotides) underlined.

**Figure 13 ijms-20-00046-f013:**
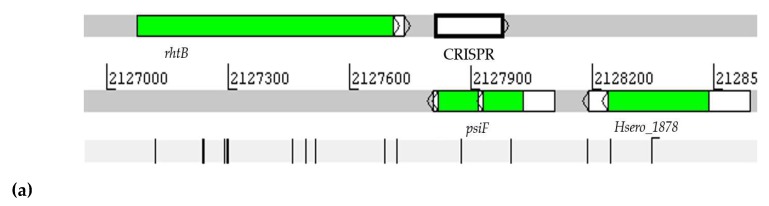
Visualization of the position of CRISPR in the genome of *H. seropedicae* SmR1 (**a**). Visualization of expression levels of CRISPR in RNA-seq assays in *H. seropedicae* SmR1. In the colored section, the 11 analyzed experimental conditions are represented, where 1 and 2 represent d.a.i (days after inoculation); (**b**) the numbers represent the value of the gene expression in RPKM.

**Table 1 ijms-20-00046-t001:** ncRNAs predicted for bacteria of the genus *Herbaspirillum* and different from those already known in *H. seropedicae* SmR1. The Infernal (version 1.1.1) computational tool was used.

Genus *Herbaspirillum*	ncRNAs
*Herbaspirillum seropedicae* Z67	ohsC_RNA, Rhizobiales-2 and UPD-PKc
*Herbaspirillum lusitanum* P6-12	Absence
*Herbaspirillum hiltneri* N3	ar14, Acido-1, Alpha_RBS, ar14, asX2, asX3, Bp2_287, csRNA, isrN, istR, Ms_AS-4, Ms_IGR-8, ncr1175, NsiR1, P26, P5, PrrB_RsmZ, RsmY, RydC, sau-5971, SpF41_sRNA, Spot_42, Xoo5, sraA, sX9, TB10Cs4H2, TB11Cs5H2 and TB9Cs1H1
*Herbaspirillum frisingense* GSF30	Entero_5_CRE, P10 and STnc430
*Herbaspirillum* spp. *estirpe* B65	snoZ248
*Herbaspirillum rubrisubalbicans* M1	ALIL, Alpha_RBS, Archaea_SRP, asX2, b55, BjrC1505, C4, ceN84, istR, K_chan_RES, MAT2A_B, MEG3_2, NRF2_IRES, OrzO-P, P36, pRNA, PtaRNA1, rli60, rox2, rseX, sau-6072, sau-63, SpF25_sRNA, SpR10_sRNA, sroB, sX15, TB9Cs1H1 and TB9Cs1H2.
*Herbaspirillum huttiense* subsp. *Putei estirpe* IAM 15032	asX2, Cardiovirus_CRE, Chloroflexi-1, MEG3_2, SBWMV2_UPD-PKk, sroB, veev_FSE, asX2, Cardiovirus_CRE, Chloroflexi-1, MEG3_2, SBWMV2_UPD-PKk, sroB and veev_FSE.
*Herbaspirillum autotrophicum* IAM 14942	MIR1444 and SpF10_sRNA.
*Herbaspirillum* spp. *estirpe* B501	Absence
*Herbaspirillum* spp. *estirpe* GW103	sX2, Flavi_CRE, RyeB, SAM-IV, sau-6072 e sroB.
*Herbaspirillum* spp. *estirpe* RV1423	psRNA2, sbcD and STnc40.
*Herbaspirillum rhizosphaerae* UMS-37	NRF2_IRES, NsiR1, Pseudomon-groES and sX9.
*Herbaspirillum rubrisubalbicans* spp. *estirpe* Os34	ryfA, SpR20_sRNA and asX2.
*Herbaspirillum rubrisubalbicans* spp. *estirpe* Os45	asX2, ROSE_2 and ryfA.

Genus *Herbaspirillum* = 15 species; ncRNAs = *non-coding RNAs*.

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
