# Peer review of "Bacterial Small RNAs in the Genus Herbaspirillum spp."

_ijms, 2018, doi:10.3390/ijms20010046_

Round 1
Reviewer 1 Report
The work presented is interesting and has no equivalent currently. Introduction brings the biological question back well. The tools and the analysis seem rigorous, but the draft is difficult to read for several points:
1- "The normalization of the sequenced samples on the SoliD platform performed by the RPKM (reads per kilobase of transcript per million mapped reads)." Perfect, but in the text, it was unclear how the experiments were performed. It is included only partially in the Methods ... strange.
2- Figures are UNREADABLE, it is simply not possible to have poor quality figures.
The reader does not want to continue when everything is so fuzzy.
3- there is no hierarchy in the questions, sections arrived after section ... what are the take home messages each time?
4- what is the interest of Figure 7. It is not really used.
5- Ausência?
6- "The ncRNAs predicted in the genus Herbaspirillum spp., Is a conservation in the sequence of the common ncRNAs, for example, Betaproteobacteria_toxic_sRNA, pfl, sX4 and alpha_tmRNA, ar35, tmRNA, among others that repeated in all species except Herbaspirillum spp. B65, Herbaspirillums pp. B501 and Herbaspirillum lusitanum P6-12 .The characterization of the ncRNAs predicted in the genus Herbaspirillum spp., Suggests that despite the difference in these species, there is sequence preservation among common ncRNAs. " The term preservention is confusing. Could you be more precise?
7- How the quality of Infernal 1.1.1 tool can be assessed here?
8- is figure 9 essential?
9- "It noted that the base pairing between Hs_noco_530 and the mRNA would occur in the RBS region of the latter." "This is a strong indication that the hs_noco_530 ncRNA negatively regulates the translation of the seryl-tRNA synthetase."
Is this a unique case?
10- There is a lack of discussion about the limitations of these approaches. Predictions and experiences have biases; it would be honest to talk about it.
Author Response
Response to Reviewer 1 Comments
Point 1: "The normalization of the sequenced samples on the SoliD platform performed by the RPKM (reads per kilobase of transcript per million mapped reads)." Perfect, but in the text, it was unclear how the experiments were performed. It is included only partially in the Methods ... strange.

Response 1: Differential expression of ncRNAs in seq RNA material, presented in the results in section 2.2 and included in the material and methods in section 4.2, were described in Supplementary Material S1 (Cultivation conditions employed in the experiments of RNAseq with H. seropedicae SmR1).
Point 2: Figures are UNREADABLE, it is simply not possible to have poor quality figures.
Response 2: Changes were made to the figures of the manuscript, (according to suggestion point 2). The resolution of the figures has been improved in both aspect of size and sharpness.
Point 3: there is no hierarchy in the questions, sections arrived after section ... what are the take home messages each time?
Response 3: The hierarchy of the sections was organized according to the content described in each section.
Point 4: what is the interest of Figure 7. It is not really used.
Response 4: We performed the change in (Figure 7), where we extracted the plot dot plot and left only analysis of the secondary structures of 6S ncRNAs, of the species: Herbaspirillum seropedicae SmR1, Janthinnobacterium sp. and Burkholderia CCGE1003. These three structures shown in Figure 7 are basically similar, but there are variations in the presence and number of loop out structures that may or may not have functional significance.
Point 5: Ausência?
Response 5: Correction of the word "Absence" was performed for "Absence", mentioned in (Table 1).
Point 6: "The ncRNAs predicted in the genus Herbaspirillum spp., Is a conservation in the sequence of the common ncRNAs, for example, Betaproteobacteria_toxic_sRNA, pfl, sX4 and alpha_tmRNA, ar35, tmRNA, among others that repeated in all species except Herbaspirillum spp. B65, Herbaspirillums pp. B501 and Herbaspirillum lusitanum P6-12 .The characterization of the ncRNAs predicted in the genus Herbaspirillum spp., Suggests that despite the difference in these species, there is sequence preservation among common ncRNAs. " The term preservention is confusing. Could you be more precise?
Response 6: A change was made: "The characterization of ncRNAs predicted in the genus Herbaspirilum spp. suggests that despite the difference of ncRNAs described from (Table 1), there are ncRNAs that are common among species".
All predicted ncRNAs of the genus Herbaspirillum spp. are described in (Supplementary Material S4), both the common ncRNAs and the deferent ncRNAs.
Point 7: How the quality of Infernal 1.1.1 tool can be assessed here?
Response 7: Adjustments were made in the text, highlighting the importance of the Infernal.1.1.1 tool to the study. This tool has the input input, the short RNA sequence database (Rfam), which through the multiple alignment covariance model, enables the result of the prediction of ncRNAs with identity. As well, it allows the integration of other data tools according to the methods to be used in the study in question. The operating system used was Linux and the command line used is in section 4.3 of the material and methods.
Point 8: is figure 9 essential?
Response 8: Figure 9 was removed because it was seen to be non-essential.
Point 9: "It noted that the base pairing between Hs_noco_530 and the mRNA would occur in the RBS region of the latter." "This is a strong indication that the hs_noco_530 ncRNA negatively regulates the translation of the seryl-tRNA synthetase. Is this a unique case?
Response 9: This is not the only case. There are other cases that were not described in the manuscript because we intend to write another manuscript emphasizing the target mRNAs, given the importance for their understanding in the regulatory process of the study bacterium.
Point 10: There is a lack of discussion about the limitations of these approaches. Predictions and experiences have biases; it would be honest to talk about it.
Response 10: It was described in section (3. Discussion), study limitations and perspective on the study.
Dear Ms. Reviewer
Thank you very much for the comments.
With kind regards,
Amanda C. Garcia
Reviewer 2 Report
There are new and useful data in this paper on ncRNAs in H. seropedicae, and as this is one of the organisms that interacts with a number of economically important crops, the work may eventually have agricultural significance. I think the paper should be published, but the manuscript badly needs rewriting, both for the english writing and the presentation of the findings; also, there no Discussion section and that needs to be added. I think the paper can not be published without a major rewriting. In addition, the manuscript is very broad and I would focus only on the cis and trans-encoded ncRNAs. What is the significance of the CRISPR findings or of the riboswitch cobalmin relative to the ncRNA roles in gene regulation or that could be related to bacterial colonization of plant roots?
Specific and minor points:
1. page 1, lines 39-43. The first ncRNA gene and transcript that was discovered and characterized was MicF RNA (Andersen et al The isolation and characterization of RNA coded by the micF gene in Escherichia coli. Nucleic Acids Res. 1987 Mar 11;15(5):2089-101.PMID: 2436145. Also, the term trans-encoded RNA was originally presented in the paper (Regulation of gene expression by trans-encoded antisense RNAs. Delihas N. Mol Microbiol. 1995 Feb;15(3):411-4. PMID: 7540245. These papers should be mentioned.
2. page 8, lines 1-4 an analysis is needed (possibly involving metabolic factors and ncRNA targets) to explain why "none of the ncRNAs were detected in H. lusitanun P6-12 and Herbaspirillum spp. strain B65." relative to the species H. seropedicae Z67, H. hiltineri N3, H. frisingense, H rubrisubalbicans M1 and H. seropedicae AU14040
3. page. 8, line 15. 6S RNA is more than a housekeeping RNA and it is also a gene regulator. It binds to σ70 RNA polymerase holoenzyme and globally regulates gene expression in response to growth in exponential phase to stationary phase. Please add reference (Wasserman KM 6S RNA, a Global Regulator of Transcription. Microbiol Spectr. 2018 May;6(3). doi: 10.1128/microbiolspec.RWR-0019-2018. PMID:29916345)
4. page 10, Fig 7 you might point out that the three structures shown in Fig 7 are basically similar (one long stem loop as shown for Fig. 7B for the 6S of Janthinobacterium spp.) but there are variations in the presence and number of looped out structures that may or may not have functional significance. Some discussion of this is needed.
The dot plots in Fig. 7 are unnecessary, I would remove them.
5. Line 11, p. 10 "Moreover, for some non-coding RNAs, such as ncHSmR1_42, there was no prediction of targets". Please explain or give some rationale why there was no prediction of targets.
6. page 11, lines 13-25. You should mention here or someplace else in the manuscript that the first experimental evidence for the binding of an ncRNA to the ribosome binding site of a messenger RNA and the blocking the binding site (Secondary structures of Escherichia coli antisense micF RNA, the 5'-end of the target ompF mRNA, and the RNA/RNA duplex. Schmidt M et al, Biochemistry. 1995;34(11):3621-3 PMID:7534474)
7. P.16, line 11 "The number of target mRNAs predicted by the TargetRNA2 tool for the H. seropedicae SmR1 ncRNAs was quite variable". Please explain.
8. P.16, line 9. "Micc", correct toMicC with a capital .
9. Some section contain wording that was not translated from Spanish, e.g., in Table 1..
Author Response
Response to Reviewer 2 Comments
Comments and Suggestions for Authors
There are new and useful data in this paper on ncRNAs in H. seropedicae, and as this is one of the organisms that interacts with a number of economically important crops, the work may eventually have agricultural significance. I think the paper should be published, but the manuscript badly needs rewriting, both for the english writing and the presentation of the findings; also, there no Discussion section and that needs to be added. I think the paper can not be published without a major rewriting. In addition, the manuscript is very broad and I would focus only on the cis and trans-encoded ncRNAs. What is the significance of the CRISPR findings or of the riboswitch cobalmin relative to the ncRNA roles in gene regulation or that could be related to bacterial colonization of plant roots?
Response: Ms. Reviewer thank you very much for the suggestions. We are happy about the manuscript. We perform the rewrite and adjustments in the structure of the sections. Her suggestion inspired us to write another article emphasizing the diversity of ncRNAs in the genus Herbaspirillum spp., Due to the breadth and importance of the subject.
We discuss the significance of CRISPR and riboswitches cobalamin, in relation to the bacterium-plant interaction listed in section 3. Discussion.
Point 1: page 1, lines 39-43. The first ncRNA gene and transcript that was discovered and characterized was MicF RNA (Andersen et al The isolation and characterization of RNA coded by the micF gene in Escherichia coli. Nucleic Acids Res. 1987 Mar 11;15(5):2089-101.PMID: 2436145. Also, the term trans-encoded RNA was originally presented in the paper (Regulation of gene expression by trans-encoded antisense RNAs. Delihas N. Mol Microbiol. 1995 Feb;15(3):411-4. PMID: 7540245. These papers should be mentioned.
Response 1The two papers suggested above were added in section 1. Introduction: end of paragraph 4 (Andersen et al., 1987) and paragraph 7 (Delihas N. Mol Microbiol, 1995).
Point 2: page 8, lines 1-4 an analysis is needed (possibly involving metabolic factors and ncRNA targets) to explain why "none of the ncRNAs were detected in H. lusitanun P6-12 and Herbaspirillum spp. strain B65." relative to the species H. seropedicae Z67, H. hiltineri N3, H. frisingense, H rubrisubalbicans M1 and H. seropedicae AU14040.
Response 2: The explanation was given "none of the ncRNAs were detected in H. lusitanun P6-12 and Herbaspirillum spp. Strain B65." In (section 2.1 of the results) in the second paragraph.
"The results are listed in (Table 1). Through this approach no ncRNAs were detected in H. lusitanun P6-12 and Herbaspirillum spp. strain B65, due to the expressive number of sacffolds and contigs that are identified in incomplete genomes. The organisms that had the highest number of ncRNAs common to H. seropedicae SmR1 were H. seropedicae Z67 H. hiltineri N3, H. frisingense, H rubrisubalbicans M1 and H. seropedicae AU14040.
Point 3: page. 8, line 15. 6S RNA is more than a housekeeping RNA and it is also a gene regulator. It binds to σ70 RNA polymerase holoenzyme and globally regulates gene expression in response to growth in exponential phase to stationary phase. Please add reference (Wasserman KM 6S RNA, a Global Regulator of Transcription. Microbiol Spectr. 2018 May;6(3). doi: 10.1128/microbiolspec.RWR-0019-2018. PMID:29916345).
Response 3: The work suggested above was added in section 2.2 ncRNAs Differential Expression in RNA-seq: end of paragraph 7 (Wasserman et al., 2018).
Point 4: page 10, Fig 7 you might point out that the three structures shown in Fig 7 are basically similar (one long stem loop as shown for Fig. 7B for the 6S of Janthinobacterium spp.) but there are variations in the presence and number of looped out structures that may or may not have functional significance. Some discussion of this is needed.
The dot plots in Fig. 7 are unnecessary, I would remove them.
Response 4: A discussion was made regarding the (Figure 7), which is included in section 2.3. Comparative analysis of the secondary structure of H. seropedicae SmR1 6S, where the dot plot graph was extracted and emphasis was given to the three secondary structures, as suggested.
Point 5: Line 11, p. 10 "Moreover, for some non-coding RNAs, such as ncHSmR1_42, there was no prediction of targets". Please explain or give some rationale why there was no prediction of targets.
Response 5: The reason was mentioned in section (3. Discussion), which appears in paragraph 10, as suggested.
Point 6: page 11, lines 13-25. You should mention here or someplace else in the manuscript that the first experimental evidence for the binding of an ncRNA to the ribosome binding site of a messenger RNA and the blocking the binding site (Secondary structures of Escherichia coli antisense micF RNA, the 5'-end of the target ompF mRNA, and the RNA/RNA duplex. Schmidt M et al, Biochemistry. 1995;34(11):3621-3 PMID: 7534474).
Response 6: The work suggested above has been added in section 3. Discussion: end of paragraph 11 (Schmidt, M. 1995).
Point 7: P.16, line 11 "The number of target mRNAs predicted by the TargetRNA2 tool for the H. seropedicae SmR1 ncRNAs was quite variable". Please explain.
Response 7: The number of target mRNAs predicted by the TargetRNA2 tool for the H. seropedicae SmR1 ncRNAs was quite variable was explained in section 3. Discussion, which appears in paragraph 8, as suggested.
Point 8: P.16, line 9. "Micc", correct toMicC with a capital.
Response 8: Correction of MicC ncRNAs was performed, as suggested.
Point 9: Some section contain wording that was not translated from Spanish, e.g., in Table 1.
Response 9: Corrections of untranslated words to English were made correctly throughout the text, as suggested.
Dear Ms. Reviewer
Thank you very much for the comments.
With kind regards,
Amanda C. Garcia
Round 2
Reviewer 1 Report
The author answered all my questions and we incorporated these modifications to the manuscript, which can be published from my point of view as it stands.
Author Response
Response to Reviewer 1 Comments
Comments and Suggestions for Authors
The author answered all my questions and we incorporated these modifications to the manuscript, which can be published from my point of view as it stands.
The manuscript was reviewed by a native English reviewer.
Other authors were added for the two reasons: 1) they are my advisers in the doctorate of the Program of Internal Medicine and Health Sciences, Federal University of Paraná; 2) Helps pay for journal value if accepted
Amanda C. Garcia1, Vera Lúcia Pereira dos Santos1, Teresa Cavalcanti2, Luiz Martins Collaço2, Hans Graf1.
1Department of Internal Medicine, Federal University of Paraná, Brazil.
2Department of Pathology, Federal University of Paraná, PR, Brazil.
Dear Ms. Reviewer
Thank you very much for the comments.
With kind regards,
Amanda C. Garcia
Reviewer 2 Report
With extensive revisions made, this manuscript has been greatly improved. On the other hand, it is far from perfect. My bottom line is that the potential agricultural importance of plant root colonization by these bacteria and a hypothetical role of regulatory RNAs in this process make the paper worthwhile publishing. With future manuscripts, I suggest the author ask a colleague to help with the writing, both for the english and the proper writing of a manuscript.
Author Response
Response to Reviewer 2 Comments
Comments and Suggestions for Authors
With extensive revisions made, this manuscript has been greatly improved. On the other hand, it is far from perfect. My bottom line is that the potential agricultural importance of plant root colonization by these bacteria and a hypothetical role of regulatory RNAs in this process make the paper worthwhile publishing. With future manuscripts, I suggest the author ask a colleague to help with the writing, both for the english and the proper writing of a manuscript.
The manuscript was reviewed by a native English reviewer.
Other authors were added for the two reasons: 1) they are my advisers in the doctorate of the Program of Internal Medicine and Health Sciences, Federal University of Paraná; 2) Helps pay for journal value if accepted.
Amanda C. Garcia1, Vera Lúcia Pereira dos Santos1, Teresa Cavalcanti2, Luiz Martins Collaço2, Hans Graf1.
1Department of Internal Medicine, Federal University of Paraná, Brazil.
2Department of Pathology, Federal University of Paraná, PR, Brazil.
Dear Ms. Reviewer
Thank you very much for the comments.
With kind regards,
Amanda C. Garcia